# Optimal Configuration of Wind-PV and Energy Storage in Large Clean Energy Bases

**Mingyi Liu** [1]**, Bin Zhang** [1]**, Jiaqi Wang** [2]**, Han Liu** [1]**, Jianxing Wang** [1]**, Chenghao Liu** [1]**, Jiahui Zhao** [1]**, Yue Sun** [1]**, Rongrong Zhai** [3] **and Yong Zhu** [1,*]

1    China Huaneng Group Clean Energy Research Institute (CERI), Beijing 102209, China;
my_liu@qny.chng.com.cn (M.L.); bin_zhang@qny.chng.com.cn (B.Z.); 19030507@emails.bjut.edu.cn (H.L.);
jx_wang@qny.chng.com.cn (J.W.); ch_liu@qny.chng.com.cn (C.L.); jh_zhao@qny.chng.com.cn (J.Z.);
y_sun@qny.chng.com.cn (Y.S.)
2    Management Committee of Beijing Future Science City, Beijing 102209, China; wlkxcgwhcyfzc@bjchp.gov.cn
3    School of Energy, Power and Mechanical Engineering, North China Electric Power University,
Beijing 102206, China; 50201839@ncepu.edu.cn
*    Correspondence: y_zhu@qny.chng.com.cn

**Abstract:** The installed capacity of energy storage in China has increased dramatically due to the national power system reform and the integration of large scale renewable energy with other sources. To support the construction of large-scale energy bases and optimizes the performance of thermal power plants, the research on the corporation mode between energy storage and thermal energy, including the optimization of energy-storage capacity and its operation in large-scale clean energy bases. In this paper, a large-scale clean energy base system is modeled with EBSILON and a capacity calculation method is established by minimizing the investment cost and energy storage capacity of the power system and constraints such as power balance, SOC, and power fluctuations. The research proposed a method of using coupled system of thermal energy storage systems primarily based on molten salt thermal storage and thermal power generation for rough modulation and using battery energy storage system for fine modulation tasks. Example of fine modulation includes frequency modulation and heating demand of the district, which significantly reduces the energy storage investment by more than 95%. A case study of a 10 MW clean energy base is conducted. The result shows that the overall pre-tax internal rate of return of the base project is 8%, which has good economic benefits.

**Keywords:** clean energy bases; capacity optimization; coordinated operation mode; power delivery tunnels; economic feasibility

## 1. Introduction

The negative impact of carbon footprint and the need for sustainability has led to increased development in clean energy such as wind and solar energy in the recent past. However, compared with traditional fossil energy, clean energy shows obvious disadvantage of intermittent and fluctuation, which limits its application in peak regulation and frequency regulation. Energy storage is an effective means to fundamentally solve above problems.

After several years of research, energy storage has shown great application value with many projects established. Mohamed Hamdi et al. [1] conducted a study on optimization of operation for a power plant operation consisting of a wind farm, solar Photovoltaic (PV) and a storage area with MATLAB and ANN. The storage area contained a Vanadium Redox Flow Battery (VRFB) and hydrogen generation and storage systems for fuel cells. The results of the study showed that optimized power plant is capable of producing a steady output to the grid at the least Levelized Cost of Electricity (LCOE), with minimum Loss of Power Supply Probability (LPSP) and excess energy. Yilin Zhu et al. [2] proposed a

two-level optimal model for hybrid electric/thermal energy storage considering Organic Rankine Cycle (ORC), which achieved an optimal battery energy storage system capacity of 1773 kWh, and a thermal energy storage system capacity of 4823 kWh, and an ORC capacity of 91.25 kW. Another study by Ahmed M. Elberry et al. [3] investigated a geological hydrogen storage technology. The LEAP-NEMO modeling toolkit was used to model Finland's electricity generation system with and without hydrogen storage. The study results showed that in both cases, $CO_2$ emissions and electricity production decreased by about 69% and fossil fuel power generation decreased by 80%. In yet another study, Emrani A et al. [4] proposed an optimal design method for the application of large-scale Gravity Energy Storage (GES) systems in a hybrid PV-wind plant, which minimizes the construction cost of GES and makes it more technically and economically competitive. Chennaif M et al. [5] developed a new extended algorithm based on Power Pinch Analysis (PPA) and the Modified Electric System Cascade Analysis (MESCA) to solve the problem of the optimal sizing of hybrid power systems with multiple energy storage facilities. Emad D et al. [6] established a generalized mathematical model with the Gray Wolf Optimizer (GWO) to determine the number of units between photovoltaic, wind and battery packs to achieve the optimal minimum target value, ultimately finding the best PV/wind/battery system size for remote locations.

In advancing clean energy, energy storage assumes a pivotal role; however, the landscape of energy storage technology presently confronts particular challenges. From the perspective of investment and maintenance, the demand on Energy Storage System (ESS) requires low capital investment, high practicability, high efficiency, long life cycle and high security. However, the capacity of single ESS has increased dramatically, leading to increased capital investment which brings high operation and maintenance risks [7]. Besides, the manufacturing process, raw material selection, and improper energy management may shorten the life cycle of an ESS [8]. From the perspective of industrial application, the energy storage is required to meet the demand of fast response and long life cycle. At present, there are few energy storage technologies can that meet multiple demands at the same time [9].

There is limited research on the optimization of large-scale integrated energy bases producing millions of kilowatts. Therefore, this paper studied the configuration of energy storage in large-scale clean energy bases and proposes a new type of optimal capacity allocation method to the participants in the base. The method proposed breaks the operational data barriers of wind power, PV power stations, and their energy storage power stations From a global perspective, and according to the power prediction, dispatching instructions, combined with the constraints of the state of SOC, capacity limitation and charging and discharging power range of the energy storage system. Take economy and power supply reliability as the objective functions, a real-time charging and discharging strategy for the energy storage system is obtained through improved optimization algorithms and the genetic algorithm, and a cooperative operation mode of wind, PV, and energy storage is formed. This research adds to the existing body of knowledge by developing a new optimal allocation method which can effectively reduce the waste of wind and PV resources, reduce the power shortage rate, and ensure the stability of a power supply. At the same time, energy storage can also be used for frequency regulation of power grids, improve the reliability of a power supply, and improve the overall power prediction accuracy of wind power and photovoltaic power stations.

## 2. Methodology

### 2.1. Optimal Configuration of Energy Storage and Multi-Form Power Sources

2.1.1. Objective Function

1. This section researched multi-form power sources and energy storage. The clean energy base is equipped with optimal wind power, PV and energy storage capacity to meet the power supply demand. According to the characteristics of each power source in the power supply system, a capacity allocation model is established with the

least investment cost and energy storage capacity of the power system, considering constraints such as power balance, SOC, and hydroelectric units. At the same time, the calculation results are evaluated in terms of load power shortage rate, charging and discharging times of energy storage and starting and stopping times of thermal power units [10,11]. Minimize the total investment cost of the system

$$minC = min(C_A + C_{OM} + C_{other}) \tag{1}$$

where $C_A$ is the total annual initial investment, $C_{OM}$ is the annual operation and maintenance cost, and $C_{other}$ is the other costs.

2.　Minimize the energy storage capacity

Power mismatch is defined as:

$$\Delta(t) = \gamma \Big[ P_{pv}(t) + P_{wind}(t) + P_{coal}(t) - P_{battery}(t) \Big] - P_L(t) \tag{2}$$

where $\gamma$ is the relative load ratio of new energy; $P_{pv}(t)$, $P_{wind}(t)$, and $P_{coal}(t)$ are the output power of PV, wind power, and thermal power at all times, respectively. If there are other power sources, they are also substituted into Equation (2) according to the output power of the above power sources; $P_L(t)$ is the load power refers to the sum of load power of the whole energy base; $P_{battery}(t)$ is the charging and discharging power of energy storage, when charging, $P_{battery}(t) > 0$, when discharging, $P_{battery}(t) < 0$. Since there are many forms of energy storage, charge and discharge can also represent the storage and release of energy in the form of heat energy.

The process of storing and releasing energy using the energy storage device can be expressed as:

$$H_{store}(t) = H_{store}(t-1) + \begin{cases} \eta_{in}\Delta(t) \\ \eta_{out}\Delta(t) \end{cases} \tag{3}$$

where $H_{store}(t)$ is the change of energy with time when the capacity of the energy storage device is not limited.

When the difference between the new energy generation and the consumption of energy storage devices is still greater than the load demand, the capacity of energy storage required at this time will increase over time. The energy storage capacity is evaluated as below:

$$E_H = \max_{T}(H_{store}(t) - \min_{i \geq t_{store}} H(t')) \tag{4}$$

The objective function to ensure the minimum energy storage capacity is as follows:

$$E_H = min\left\{ \max(H_{store}(t) - \min_{i \geq t_{store}} H(t')) \right\} \tag{5}$$

### 2.1.2. Constraints

1.　Instantaneous power constraint [12,13]

The amount of daily energy imbalance is defined as below:

$$W_i = E_{battery}(i-1)\eta_{in} + E_{wind}(i) + E_{pv}(i) + E_{coal}(i) - Q_L(i) \tag{6}$$

where $E_{battery}(i-1)$ is the surplus electricity stored in the previous day; $\eta_{in}$ is the charging efficiency of battery; $E_{wind}(i)$, $E_{pv}(i)$ and $E_{coal}(i)$ are the total power generation of wind, PV and thermal power on that day, respectively; $Q_L(i)$ is the total load of the day.

When $W_i > 0$, it means that the sum of the four power sources of wind power, PV, thermal power, and energy storage can meet the load demand. At this time, there is still a part of the electricity in the storage battery, and the system does not need to perform load-shedding operations. When $W_i < 0$, it means that the sum of wind, PV, thermal power, and energy storage output does not meet the load demand today, and the system can only

be maintained by performing load-shedding operations. In this case, the load-shedding power is $|W_i|$, and the residual energy of energy storage is 0.

2. Energy storage system state of charge constraint [14–16]

$$SOC = H_C(t) / E_H \tag{7}$$

where *SOC* is the constraint value of the state of charge of the energy storage system; $H_C(t)$ is the actual storage capacity of the energy storage device.

Since the stored power will increase over time, the change of the stored capacity with time under constraints can be obtained with Equation (3). When configuring the power supply capacity of the base, wind power, photovoltaic power, and thermal power should meet the power supply requirements of the load as much as possible, so as to avoid repeated charge and discharge of energy storage that will affect battery life and system investment costs.

3. Output fluctuation constraint [17,18]

Energy storage is mainly used to smooth the total output power of wind and PV. Using the energy management system, the total output value and the reference output value of wind, PV, thermal power, and energy storage can be known. The fluctuation constraint is to make the difference between the former and the latter not exceed the allowable value, which can be expressed as:

$$\delta = \frac{\sum\limits_{t=0}^{T} time(|P_{pv-wind-coal-battery}(t) - P_L(t)| \geq \Delta P_{\max}}{T} \leq \delta_L \tag{8}$$

where $\Delta P_{\max}$ is the maximum fluctuating power; $\delta_L$ is the probability that the maximum fluctuating power exceeds the allowable value.

### 2.1.3. Evaluation Indexes

As a commercial clean energy base, in addition to adhering to green development, the profitability of the overall base is an important indicator for assessing the suitability of multiple sources and storage bases. This part evaluated the advantages and disadvantages of the optimal configuration of the system based on technical economics.

1. Static indexes [19–21]

Static indexes refer to the indexes that directly calculate the cash flow formed using investment projects without uniform conversion according to the time value of money, including static investment pay-back period and investment profit rate, etc.

Static Pay-Back Period (SPP) refers to the static pay-back period during the construction period and the static pay-back period excluding the construction period.

$$\sum_{n=1}^{PP} (CI_n - CO_n) = 0 \tag{9}$$

where *PP* is the static pay-back period; $CI_n$ is the cash inflow in the nth year; $CO_n$ is the cash outflow in the nth year.

Return on Investment (ROI) refers to the ratio between the annual profit (earnings before interest and tax total) and the total investment of the project in normal productive years.

$$ROI = EBIT / TI \tag{10}$$

where EBIT is the annual profit of the project; TI is the total investment of the project.

The Rate of Return on Common Stockholders' Equity (ROE) refers to the ratio between the annual after-tax net profit and the total capital of the project in normal productive years.

$$ROE = NP/EC \tag{11}$$

where $NP$ is the after-tax net profit of the project; $EC$ is the total capital of the project.

2.  Dynamic indexes [19,22,23]

Dynamic indexes refer to the indexes that uniformly convert the cash flow formed using investment projects according to the time value of money.

Dynamic Pay-Back Period (DPP) refers to the time required to recover the initial investment when considering the time value of money.

$$\sum_{n=1}^{Pt} \frac{(CI_n - CO_n)}{(1 + BY)^n} = 0 \tag{12}$$

where $Pt$ is the dynamic pay-back period; $BY$ is the benchmark yield.

Net Present Value (NPV) refers to the sum of the net cash flow of each year in the project life cycle converted to the present value of the starting point of the development project according to the benchmark rate of return of the industry, and it is calculated as follow:

$$NPV = \sum_{n=1}^{N} \frac{C_n}{(1 + BY)^n} \tag{13}$$

where $C_n$ is the annual net cash flow of the nth year.

Annual net cash flow refers to the difference between the annual cash inflows and the annual cash outflows in the current year, namely:

$$C_n = CI_n - CO_n \tag{14}$$

Internal Rate of Return (IRR) refers to the discount rate when the total present value of capital inflow is equal to the total present value of capital flow, and it is calculated as follows:

$$NPV = \sum_{N=1}^{N} \frac{C_n}{(1 + IRR)^n} = 0 \tag{15}$$

Levelized Cost of Electricity (LCOE) refers to the ratio of all costs and total power generation during the operation period of a power generation project, namely:

$$LCOE = \left( \sum_{n=0}^{N} \frac{Cost_n}{(1 + r_{dis})^n} \right) \bigg/ \left( \sum_{n=0}^{N} \frac{P_{gen,n}}{(1 + r_{dis})^n} \right) \tag{16}$$

where $Cost_n$ is the cost of the project in the nth year; $P_{gen,n}$ is the power generation of the power station in the nth year.

In the numerator of the above equation, the annual operation cost of the power station is converted, but the calculation time starts from the 0th year, and this part of the investment does not need to be converted. Therefore, the Equation (16) can be written as:

$$LCOE = \left( I_{cc} + \sum_{n=1}^{N} \frac{Cost_{ann,n}}{(1 + r_{dis})^n} \right) \bigg/ \left( \sum_{n=1}^{N} \frac{P_{gen,n}}{(1 + r_{dis})^n} \right) \tag{17}$$

where $Cost_{ann,n}$ is the cost of the nth year of the project, including operation and maintenance costs, fuel costs, insurance costs, etc.

*2.2. Strategy Optimization of Energy Storage and Multi-Form Power Sources Operation*

2.2.1. Coupling Mode between Energy Storage and Multi-Form Power Sources

The energy base system includes power sources such as wind power, PV, and thermal power while energy storage include battery energy storage, heat storage, and hydrogen energy, as well as heating, electricity, cooling, and gas. The coupling modes among the main power in the system are more complicated and the connection modes are more diverse. Coupling power sources and energy storage is designed according to the energy form, that is, prioritize the construction of the main connection network according to the main electric energy transmission. Secondly, connection lines are added according to the order of energy storage, and finally attribute the same type of energy to the same external export according to the load demand.

1.　　Main power connection transmission network

The main electric energy includes wind power, PV, and thermal power, and the installed capacity of wind power and photovoltaic accounts for the main part. Therefore, when designing the coupling mode, wind power and photovoltaic are regarded as the main power. According to the distribution position of power stations and the thermal power station sites, the shortest path algorithm which is used to optimize the best center point are used as a collection site for energy bases for electricity transmission.

There is already a lot of work on the shortest path algorithm, including the depth- or breadth-first search algorithm, Floyd algorithm, Dijkstra algorithm, Bellman–Ford algorithm, etc.

2.　　Energy storage connected to the main grid

There are many energy forms which are divided into two based on the geographical perspective location requirements. One type requires high geographical location, such as pump hydro storage, compressed air storage, etc., and the other one does not require high geographical location, such as thermal storage and battery energy storage.

When optimizing the location of energy storage and designing the connection between energy storage and the main power transmission network, priority should be given to the energy storage forms with high geographical requirements. Additionally, with the above center point as the center, based on the surrounding geographic time, the energy storage that does not require high geographic location is arranged systematically for easy acess.

3.　　Load distribution network

The characteristics of each user should be considered when designing the load distribution network due to load diversity. If the electricity users are in the Ultra High Voltage (UHV) receiving market, they only need to set up the collection station as mentioned above. Gas users (such as hydrogen) are generally chemical plants or central cities, so when selecting hydrogen production stations, hydrogen production stations can be selected in the vicinity of chemical plants or central cities to replace hydrogen transmission in the form of power transmission. Generally, heating load and cooling load can be obtained using electric energy. Because of the limitation of long-distance transmission of cooling and heating load, it can be built near the load center.

2.2.2. Coordinated Operation Mode between Energy Storage and Multi-Form
Power Sources

1.　　Coordinated operation strategies between multi-form power sources and energy storage

(1) New energy priority grid connection

For the clean energy base, wind power and photovoltaic power generation are the mainstay. When the base is in a coordinated operation, priority is given to the grid connection of new energy generation to reduce carbon emissions.

(2) Deep coupling of thermal power and heat storage

Thermal power has the characteristics of large capacity, and its deep peak regulation ability is strong. Thermal storage has the advantages of large capacity and low cost and

is easy to be deeply coupled with thermal power (furnace side, machine side). Therefore, thermal storage systems can be regarded as an important energy storage method. First, it stores excess renewable electric energy. Futher, it provides an auxiliary "boiler" for thermal power. When the thermal storage capacity is enough to support the deep adjustment of thermal power units, the thermal storage energy is given priority, that is, the renewable energy is given priority. Otherwise, the boiler is turned on to provide some energy.

The thermal storage material should be molten salt whose temperature range matches that of thermal power, and its composition is 60% sodium nitrate and 40% potassium nitrate.

(3) Flexible energy storage fine adjustment

Flexible energy storage (such as batteries) tends to have strong, fast adjustment capabilities, but its cost is more expensive than thermal storage. Therefore, this type of energy storage is only used for fine regulation, such as frequency regulation.

2.  Energy-based scheduling strategy

For the energy base, because there are many power sources and energy storage subjects, the system structure is complex. Therefore, the scheduling strategy should be simple and direct, and the insufficient part should be compensated using fine regulation using flexible energy storage. Its main scheduling strategy is as follows:

(1) The load of wind power and photovoltaic power generation is higher than the power demand

Wind power and photovoltaic are directly connected to the grid according to the electricity demand. The surplus electricity is mainly used to heat the thermal storage medium and store it in the battery which is used for fine adjustment.

(2) The load of wind power and photovoltaic power generation shall not be higher than the power demand

The wind and PV power should match power demand without exceeding it. Both wind and solar connect to the grid; thermal sources like storage and coal fill any deficit. Thermal power relies on storage and coal. Sufficient storage means storage alone provides heat, and the boiler stops. Insufficient storage makes both provide heat. The battery energy storage is used for fine adjustment in this process.

(3) Other requirements

When heating is needed, the thermal energy storage system supplies the required warmth. Hydrogen is generated from electricity through water electrolysis when needed. The surplus electricity generated by wind power and PV power generation is preferentially used to produce hydrogen. If there's excess capacity in the thermal power load (currently operating at 80% load, leaving 20% as surplus), it's utilized to produce hydrogen using the extra electric energy.

## 3. Structure of Clean Energy Base

### 3.1. System Structure

The clean energy base includes wind power, PV, and thermal power in terms of the source side and it connect the power grid of the external power transmission landing point. The load side includes heating load, cooling load, and gas etc. In terms of the storage side, there are various energy storage forms, such as battery, hydrogen, thermal energy storages, etc., as shown in Figure 1.

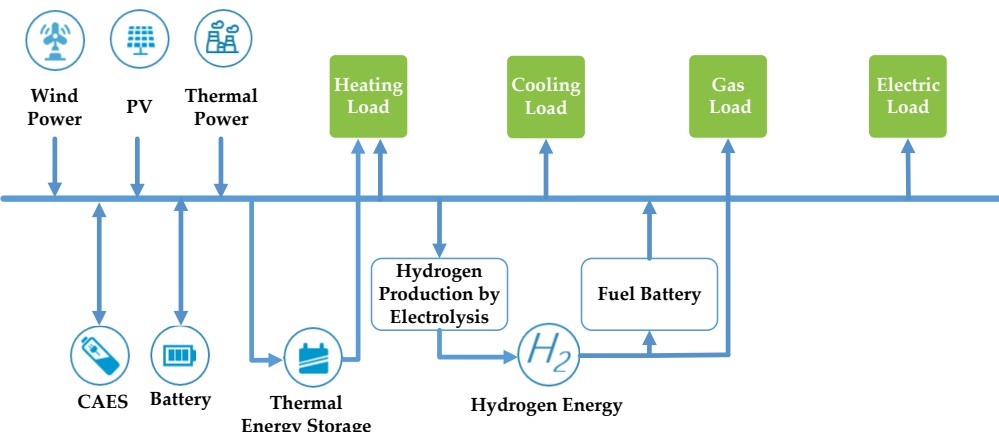

**Figure 1.** Structure of clean energy base.

*3.2. Power Characteristics*

3.2.1. Research on Wind Power Characteristics

Due to the fluctuation of wind speed, the output of wind power is also random, fluctuating, and intermittent.

The wind power output exhibits substantial fluctuation amplitudes, accompanied by irregular frequency oscillations. In extreme cases, the daily output active power value of wind farms may change continuously within the range of 0–100%. As shown in Figure 2, the minimum value closely approximates zero, while the maximum value exhibits to attain or even slightly surpass the rated capacity of 1188 MW. The daily average power generation of the same wind farm on adjacent days may be similar or the same, but the output difference of each period of wind power generation is very obvious. It shows that even though the wind turbine can obtain better wind energy utilization coefficient through advanced control technology, the active power output of wind turbine will still present random instability based on the inherent uncertainty of wind energy.

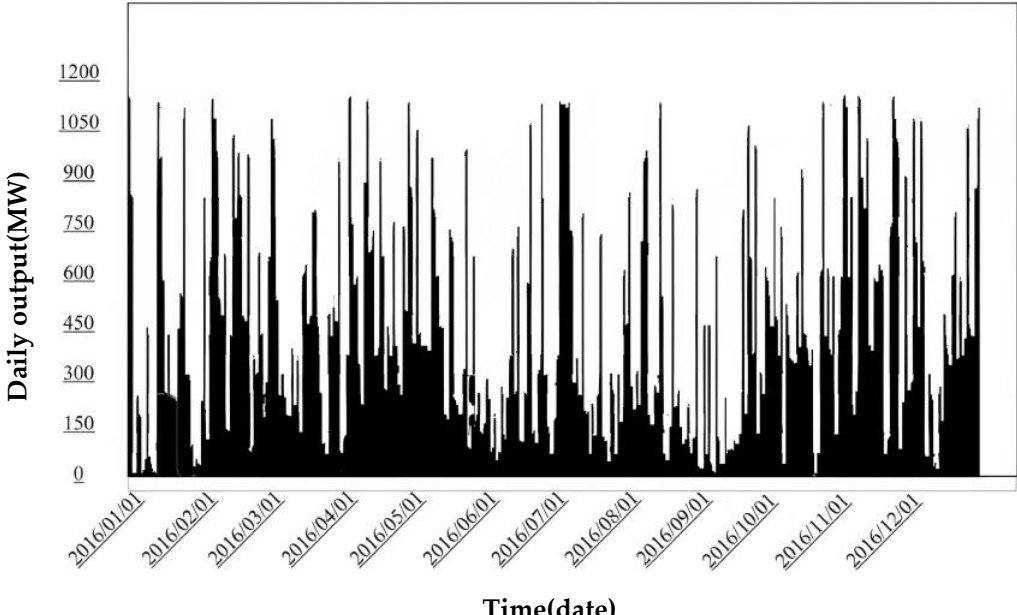

**Figure 2.** The daily output of wind farm.

Wind power generation is not only affected by both short-term and long fluctuations. It is also affected by factors such as seasonal changes and climate changes [24]. The wind farm can be reasonably arranged for maintenance according to the output characteristics of wind

power [25]. Determining the fluctuation type of daily output of wind power generation is of great significance to the planning, scheduling and operation of peak shaving and standby of the power grid with wind power generation.

### 3.2.2. Research on PV Characteristics

The evident characteristics of PV output encompass pronounced randomness and intermittent behavior. The output characteristics is not only be affected by environmental conditions such as solar illumination, but also by time, geographical area, and other related factors. If the PV power penetration rate of the low-voltage distribution network increases, most of the external meteorological conditions in the adjacent areas of this distribution network are basically the same. When connected to the distribution network, a large number of distributed PV power sources and loads in this area are related to each other, and the photovoltaic output will increase or decrease at the same time, which will make the photovoltaic output more random and produce greater fluctuations.

With the continuous development of technology, many researchers began to study the distributed photovoltaic access distribution network. As the scale of photovoltaic grid integration steadily expands, the influence of photovoltaic power generation characteristics on the distribution network becomes increasingly conspicuous. Consequently, ensuring the stability of the distribution network necessitates a greater consideration of control factors [24,26].

### 3.2.3. Research on Thermal Power Characteristics

Thermal power generation includes various forms, such as coal-fired power generation and gas-fired power generation. Coal-fired power generation is studied in this paper. The fuel, air, and water (once-through boiler) regulating system of a unit controlled using a coordinated control system can be regarded as the follow-up system of boiler instructions. The load regulation performance on the boiler side can be simplified to the response characteristics of steam heat output to boiler instructions.

For steam, the adjustment quantity related to the load is mainly the adjustment of the steam turbine, which quickly follows the change of the turbine command, and the change of the steam flow and pressure caused by the change can be regarded as a relatively fast inertial link.

The load change performance of coal-fired units mainly depends on the response characteristics of the load to the turbine governor and boiler combustion rate while considering the change of main steam pressure. When the throttle is changed, even if the firing rate is unchanged, the heat storage of the boiler can make the load change quickly and maintain the general time. The heat storage capacity of the boiler mainly depends on the type of boiler. The heat storage capacity of a drum boiler is larger than that of a once-through boiler. Depending on the capacity of the steam drum or header and the size of the heating surface of the boiler. In addition, its size is related to the main steam pressure. The high main steam pressure has a strong heat storage capacity, while the low main steam pressure has a weak heat storage capacity. For the drum boiler, the heat storage of the boiler is better expressed using the change of drum pressure signal.

The conversion of steam heat into electric load is an extraordinary process and the response characteristics of unit load to boiler instructions mainly depend on the boiler response characteristics, which is a high-order inertia link with relatively large delay and is related to the type of pulverizing system and boiler.

### 3.2.4. Energy Storage

1. Battery energy storage

Battery energy storage is the conversion of electrical energy and chemical energy using electrochemical reactions to realize the storage and release of electrical energy [27]. Battery energy storage offers versatile installation and rapid response, playing a crucial role in delivering essential power and energy services. The most common use of battery energy

storage is frequency regulation, followed by spare capacity, electricity fee management, and energy time migration [28].

2.　　Thermal energy storage

At present, molten salt heat storage technology is the most widely used heat storage technology. Molten salt thermal storage is based on the characteristics of a large temperature range, high specific heat capacity, and good heat transfer performance of molten salt [29]. Currently, this technology is mainly used in solar thermal power stations. At the same time, it can also participate in new energy generation consumption and deep peak shaving of thermal power units, heating, etc.

The main advantages of using molten salt heat storage to store energy are large scale, the scale of energy storage depends on the capacity of molten salt, and molten salt materials are easy to obtain; it can be conveniently used with conventional coal-fired/gas-fired units, and high-temperature molten salt can replace the boilers of conventional coal-fired/gas-fired power stations to provide heat and generate electricity with the help of the existing power generation system. The main disadvantages are: the application is limited by the development and application of solar thermal power generation technology; the demand for heat preservation and condensation prevention is substantial, and since the molten salt tank and pipeline need to be insulated and anti-condensate during system operation, it causes large energy consumption [30].

### 3.2.5. Hydrogen Energy

Hydrogen energy utilization involves four processes: hydrogen production, hydrogen storage, hydrogen transmission, and hydrogen utilization. In recent past, demonstration projects with new energy power generation that electrolyze water to produce hydrogen have been carried out. Electrolyzed water is a high-energy-consuming hydrogen production method [31]. Using new methods to generate electricity when the grid load is low is one of the ways to improve the utilization rate of new energy generation. The energy density of hydrogen in the standard state is only 8.4 MJ/L. New energy power generation and hydrogen storage technology involve the electrolysis of water, a process wherein water is decomposed into hydrogen and oxygen, effectuating the conversion of electrical energy into chemical energy [32].

At present, the electrolytic hydrogen production technology consumes 4.42–5.5 kWh of electricity to produce standard hydrogen [33].

The advantages of hydrogen energy storage technology include the following: the calorific value per unit weight is $1.4 \times 10^8$ J, which is the highest except for nuclear fuel; no pollutants are generated; large-scale energy storage can be realized. The main disadvantages include presently low energy conversion efficiency, high cost and large infrastructure investment, safety concerns, etc.

## 4. Case Study

### 4.1. System Modeling

This section researched a certain 10-million-kilowatt clean energy base to build a system model. Combined with the actual situation of the base, a case study is conducted with the method described in Section 2.

### 4.1.1. Wind Power Generation System Model

A 10-million-kilowatt clean energy base is rich in wind energy resources, with a wind speed of about 5 m/s–9 m/s at a height of 90 m, which has great development potential. According to the resource characteristics and terrain conditions of the planning area, firstly, the wind turbines with better power generation capacity are mainly concentrated in the models with a single capacity of 2.5–3.3 MW. Secondly, on the basis of fully considering the safety of the wind turbines, it is suitable to choose a wind turbine with a larger impeller diameter. Finally, the power generation capacity of the model can be more intuitively reflected from the unit kilowatt sweeping area. The unit kilowatt sweeping area of the top

three models is about 6 square meters/kilowatt. Due to the wide geographical distribution of the wind farm, the wind turbines are partitioned when modeling, and the wind resource data of each partition point is selected as the meteorological data input for separate modeling. The wind power generation system model shown in Figure 3 was built on EBSILON® Professional [34].

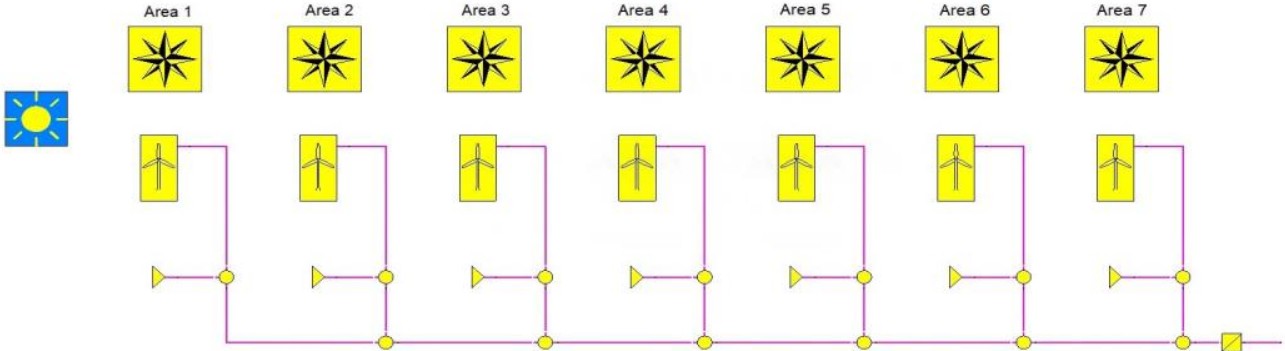

**Figure 3.** Wind power generation system model.

### 4.1.2. PV Power Generation System Model

The base is one of the areas with abundant solar energy resources, with annual sunshine hours of 2800–3200 h, a sunshine rate of 64–73%, and a frost-free period of 110–130 days. The overall distribution of solar energy resources is uniform, and the total solar radiation value in most areas can reach 6000 MJ/m$^2$, which is suitable for the development and construction of solar power generation projects. The PV power generation system model is shown in Figure 4.

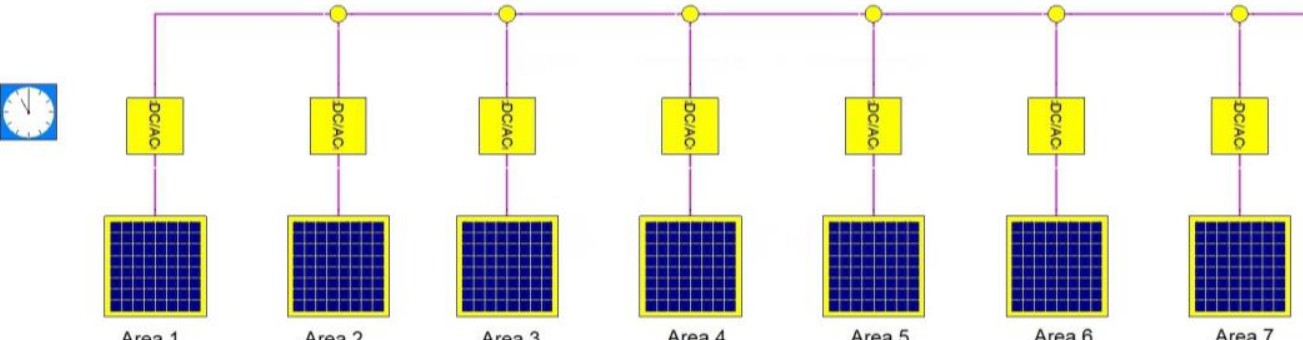

**Figure 4.** PV power generation system model.

### 4.1.3. Thermal Power Generation System Model

Considering the current power situation of the country, the power grid structure of the base, the power grid system, and other factors, it is recommended to use thermal power as a support for new energy power generation. After long-term development, thermal power generation has been completed and matured, and the million-kilowatt thermal power with high parameters, large capacity, and high performance will become the main type of thermal power growth in the future. Therefore, the million-kilowatt thermal power unit is mainly studied in this paper, and its system model is shown in Figure 5 shown.

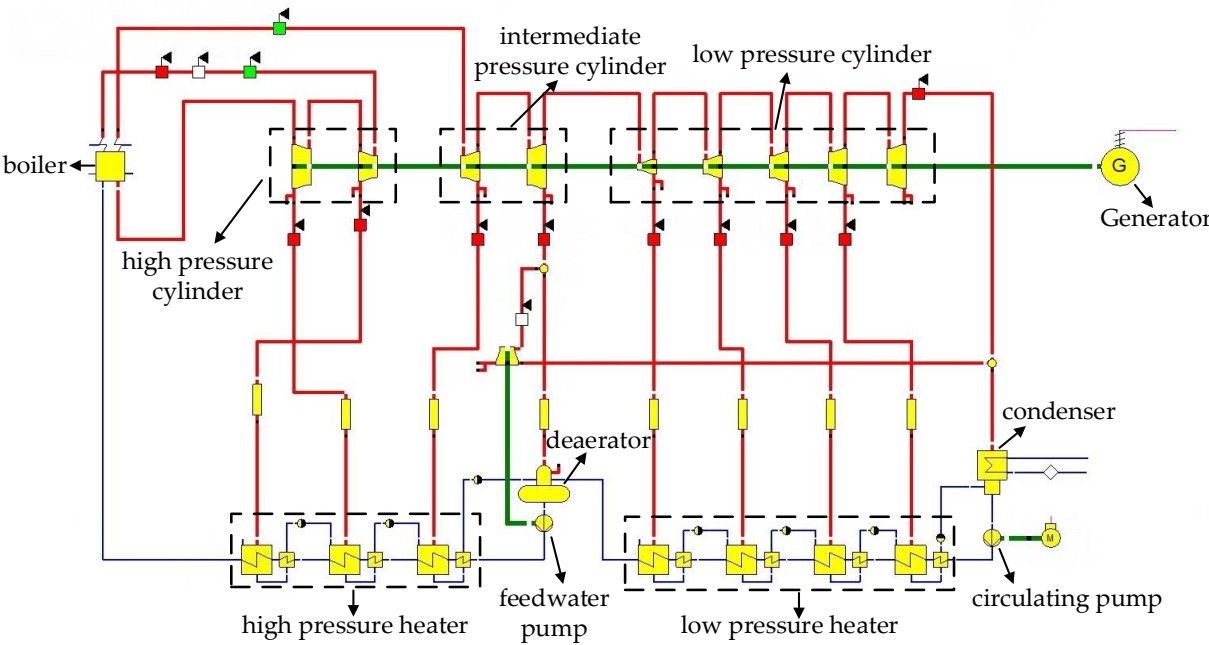

**Figure 5.** Thermal power generation system model.

### 4.1.4. Battery Energy Storage System Model

At present, the new energy base is connected to the grid, and a large number of rapid power fluctuations will affect the stable operation of the receiving grid. The receiving power grid not only needs to provide more frequency regulation capacity but also puts forward higher requirements for frequency regulation response time. The most economical and effective way to develop new energy in the future is to configure an energy storage system with certain power in the wind farm to suppress short-term large wind power fluctuations, realize the tracking of dispatching curves, and solve the problems of wind farm side, channel, and receiver side with the characteristics of fast response of energy storage. Among various forms of energy storage, battery energy storage is the most flexible, and its system model is shown in Figure 6:

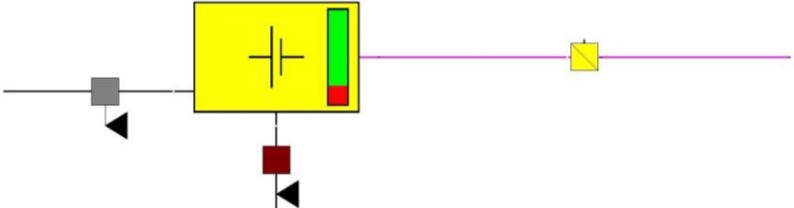

**Figure 6.** Battery energy storage system model.

### 4.1.5. Hydrogen Energy System Model

If the new energy base encounters a surplus of electricity or has a good hydrogen energy utilization market, it can realize electricity-hydrogen energy conversion using on-site or off-site hydrogen production. When hydrogen is produced on-site, the electric energy does not pass through the power grid. So, the hydrogen production project has the advantage of electricity price cost. When hydrogen is produced off-site, the power transmission and distribution price and electricity price surcharges will be added after the hydrogen production electricity is connected to the grid at a parity price, which makes the cost of hydrogen production electricity higher. The hydrogen energy system model is shown in Figure 7 below. "Hydrogen production equipment" is electrolyzed water hydrogen production equipment. After the electricity from the wind farm is input into the equipment, hydrogen can be electrolyzed. When designing, it is calculated that 5 kWh of

electricity can be electrolyzed to produce one standard cubic meter of hydrogen. Due to the small scale of electrolyzed water hydrogen production equipment, the fluctuation of wind power output has little impact on it, so the fluctuation in the electrolysis process is not considered.

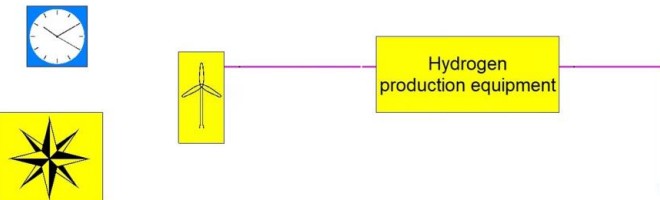

**Figure 7.** Hydrogen energy system model (hydrogen production from wind power).

Combine the above models organically to form the overall model of the base. The outgoing transmission line is the main line, which connects the main power, such as wind power, photovoltaic, thermal power, and energy storage. The details of each main power are as mentioned above. The power load demand of the power grid can be set at the outgoing line. Combined with the current situation of wind, PV, thermal power and energy storage, each connecting subject operates in an orderly manner according to the optimized dispatching mode.

### *4.2. Power Demand and Generation Characteristics*

#### 4.2.1. Power Demand

This paper researched the hourly electricity load of a certain province. According to the power generation capacity of the energy base, the maximum load of 10 million kilowatts is used for conversion. As can be seen from Figure 8, the annual electricity load changes greatly, with the highest load of 10 million kilowatts, which is three times the lowest load (3.269 million kilowatts). It is extremely difficult to track the electricity load demand, which depends on the power generation situation of new energy sources. Analyze the hourly load data and count the occurrence frequency of each load in a year (Figure 9), it can be found that the load frequency of 8760 h in the whole year presents a nearly normal distribution, and its expected value is close to 6600 MW. Therefore, in the follow-up system optimization process, this expected value should be used as the basis for power generation design.

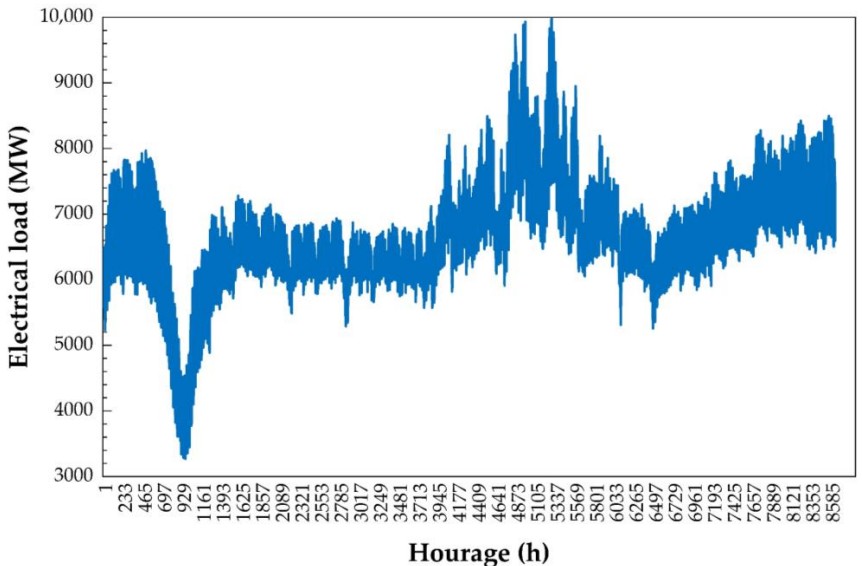

**Figure 8.** Hourly power consumption load of a province throughout the year (converted to a maximum of 10 million kilowatts).

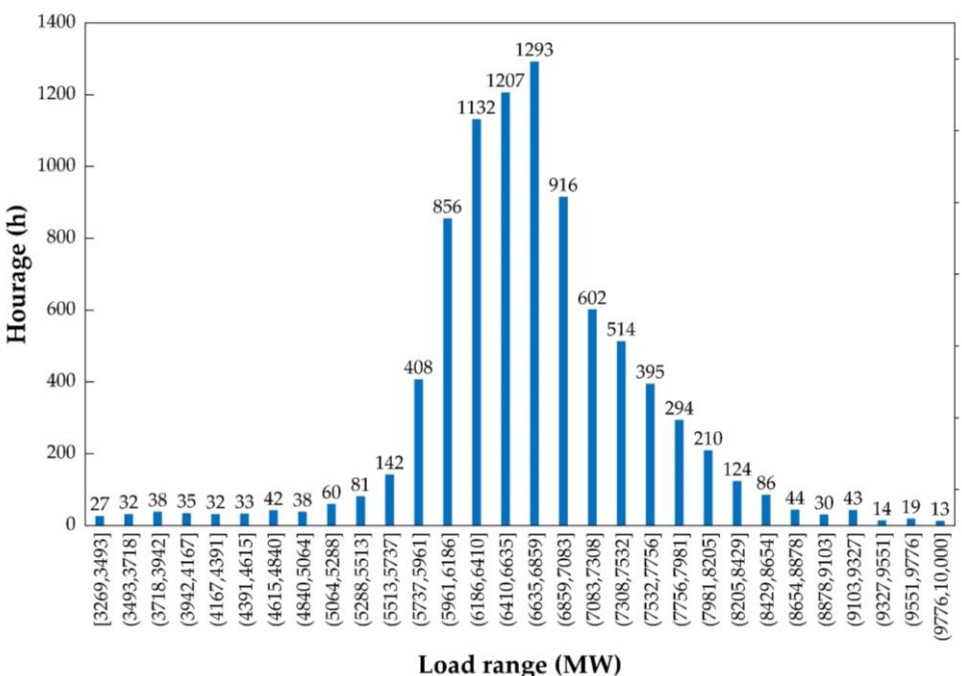

**Figure 9.** Distribution of hourly electricity load in a province throughout the year.

### 4.2.2. Generation Characteristics

1. Characteristics of new energy power generation

For a clean energy base, wind power and PV are the main power of new energy, so it is necessary to ensure that this part of electricity can be connected to the grid as much as possible to maximize clean power generation. Therefore, in this part, the superposition of the two power generation loads is considered, the hourly load data were analyzed, and the frequency of each load in a year is counted (Figure 10). It can be found that the load frequency of 8760 h throughout the year is basically stable, and the load range of 5300 MW has the most hours. So, this load is the power generation peak that should be paid attention to when designing the energy base. When thermal power is configured for adjustment, the high load of this part is mainly adjusted.

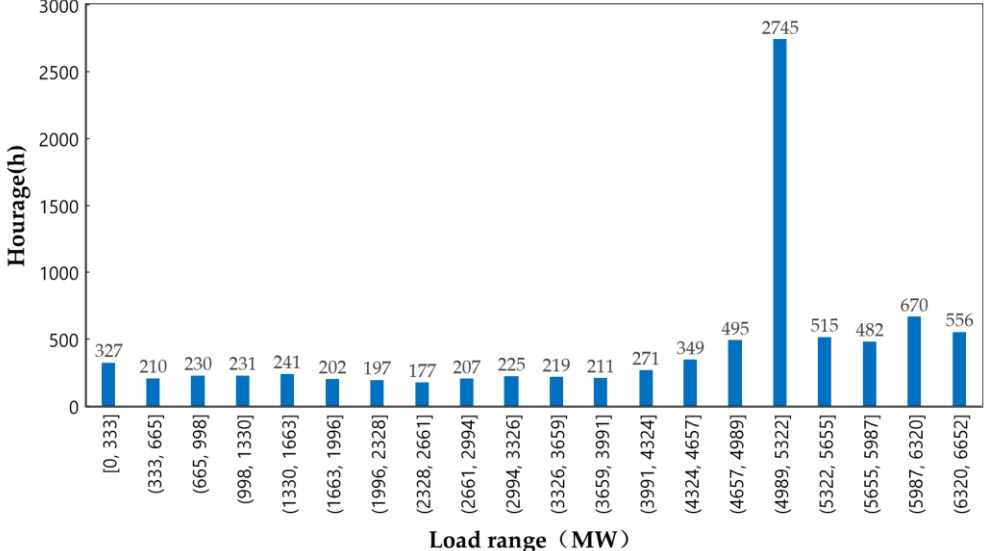

**Figure 10.** Load distribution of wind power and PV power generation hourly(superposition).

2. Characteristics of thermal power generation

To support the safe and stable operation of the energy base dominated by new energy sources according to the demand of power dispatching, it is necessary to allocate a certain amount of thermal power. The allocation principle is to ensure the stable operation of the energy base, and secondly to ensure the annual utilization hours of thermal power (to ensure economy) and reduce frequent startup (to ensure safety). At present, thermal power units can basically achieve 40–100%THA regulations, but the regulation process is delayed.

Figure 11 shows the demand for energy base for adjustable power sources (electricity load demand minus wind power and photovoltaic power generation load). It can be seen from the figure that according to the current electricity load and wind and photovoltaic power generation capacity, the energy base has a greater demand for adjustable power sources. The maximum demand for thermal power support at a single point (an hour) is 9280 MW, and the average demand is 2575 MW. The maximum surplus of wind power generation at a single point is 2851 MW, and the average surplus is 730 MW. Therefore, when signing a power transmission agreement with the UHV receiving power grid, it is impossible to follow the actual load curve of up to 10 million kilowatts, because the wind and PV main power sources cannot meet the power demand. Also, there is a huge demand for thermal power. It is also advisable to provide electric energy completely according to the local power load demand curve, otherwise the demand for energy base regulation is higher.

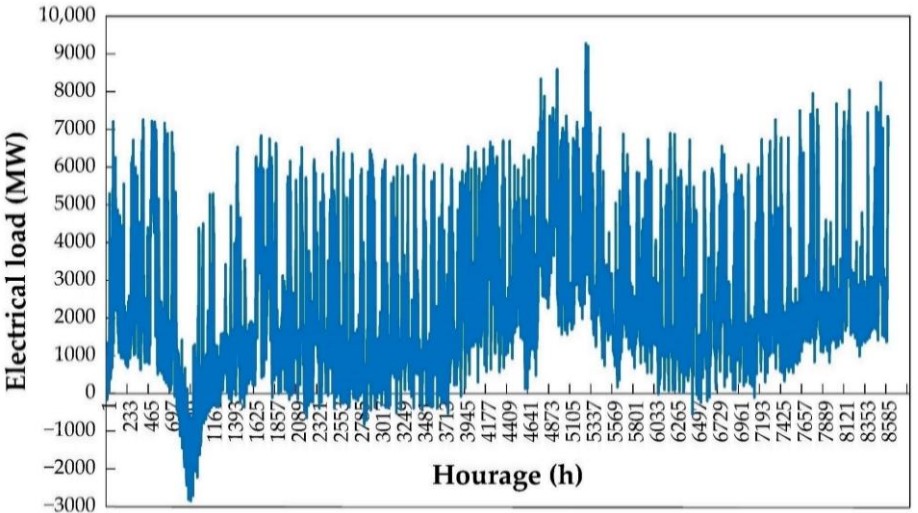

**Figure 11.** Demand for adjustable power supply in energy base (electricity load demand minus wind power and photovoltaic power generation load).

*4.3. Reverse Design with the Goal of Stable Output*

From the above research and Figure 10, it can be found that the load frequency of wind and PV power generation for 8760 h throughout the year is basically stable, and the number of hours in the load range of 5000 MW to 5300 MW is the highest. The power generation with a load of less than 3500 MW accounts for 10% of the total annual power generation; the low-load points are mainly distributed at night and in the morning (21:00–9:00 the next day). Similarly, the power generation with a load of more than 3500 MW accounts for 90% of the total annual power generation, and the high load points are mainly distributed in the daytime and evening (10:00–20:00).

Figure 12 intuitively shows the average power generation load of wind power and PV at each time point. The average power generation load of wind power and PV varies greatly during the day and night. Therefore, to ensure the economy of energy base development, power sale agreement should not be signed according to the actual load curve of the power grid, but should aim at stabilizing the output, reverse design and optimize the

configuration of the base, and design the power transmission load separately according to different time periods.

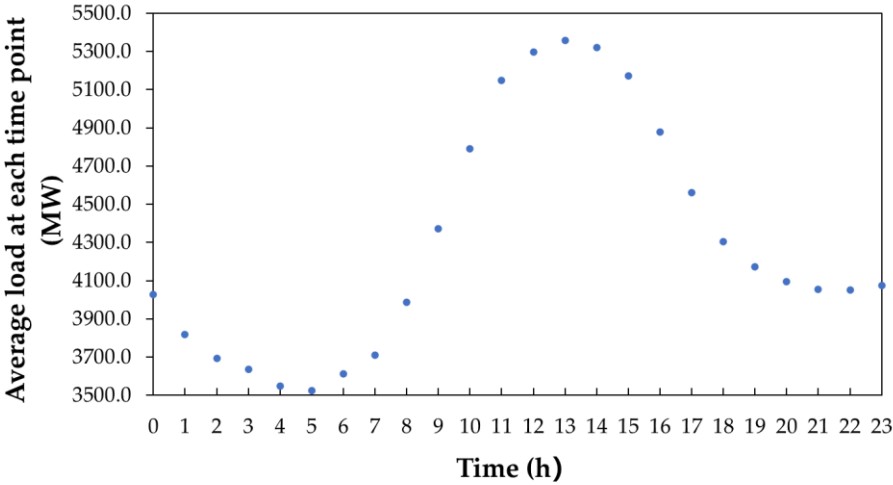

**Figure 12.** Average power generation load of wind power and photovoltaic at each time point.

### 4.4. Results and Discussion

#### 4.4.1. Optimal Configuration Scheme

According to Figure 12, the average load is taken as the basis of the load of the energy base, and thermal power or energy storage support is considered appropriate. Figure 13 shows the utilization of heat storage with different thermal storage capacities. It can be seen from the figure that the maximum heat storage required by the energy base is 371 GWh electricity equivalent, which means all excess power generated can be stored in the heat storage container at this time. Reducing the heat storage capacity will lead to the fact that part of the surplus electric energy cannot be stored in the heater, and the utilization rate of the surplus electric energy (relative to the maximum heat storage capacity) will gradually decrease from 100%. When the heat storage capacity is between 100 GWh and the maximum capacity, the decrease rate is gentle. When the thermal storage capacity is below 100 GWh, the decrease in thermal storage utilization rate is accelerated. Based on a comprehensive evaluation, the storage heat capacity of 80 GWh is taken, and the utilization rate of excess electric energy is 85.4%.

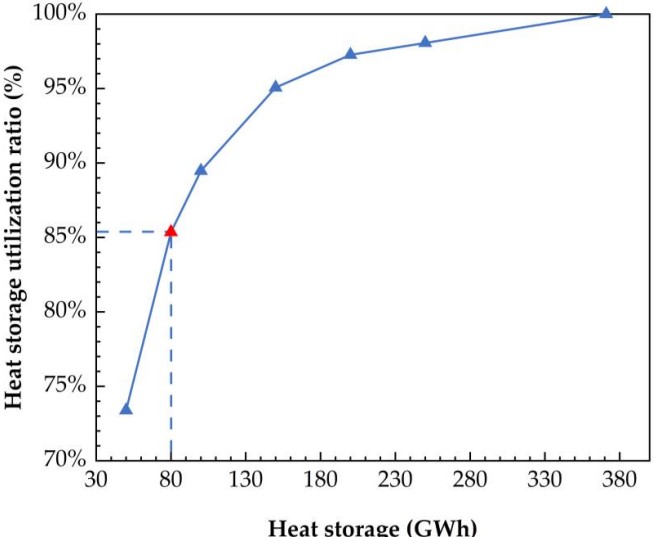

**Figure 13.** Utilization of heat storage by different heat storage capacities.

The above values do not affect the demand for energy base for flexible and small-capacity energy storage. According to the calculation, the energy base needs to discharge 46.8 GWh of flexible and small-capacity energy storage annually. Based on the required operating hours (325 h), the average discharge power is 144 MW, and the required time is 1 h. The battery energy storage system can meet the above operation requirements. The required electric energy is provided by excess wind power and PV power.

Combined with the above analysis, the optimal configuration scheme of clean energy storage and multi-form power sources is 10 million kilowatts for wind power, 2 million kilowatts for photovoltaic power, and 4 × 1 million kilowatts for thermal power. After comprehensive optimization and calculation, 80 million kWh of molten salt heat storage capacity is required, which can solve 85.4% of the power rationing problem. At the same time, it is necessary to allocate an average of 144,000 kW (1 h) of electrochemical energy storage, which can flexibly solve the fine adjustment required by the energy base.

The power generation composition of the clean energy base is shown in Figure 14. Within the total electricity generation of 37.72 billion kWh, wind power and PV contribute significantly with 34.14 billion kWh, constituting a noteworthy 90.5%. Conversely, coal-fired power generation contributes 3.58 billion kWh, representing 9.5% of the aggregate energy production. Among wind power and PV power generation, the direct power generation is 31.42 billion kWh, accounting for 92.1%; the indirect power generation by thermal storage is 2.67 billion kWh, accounting for 7.1%; the indirect power generation by battery storage is 50 million kWh, accounting for 0.1%. The total amount of energy storage power generation is 2.71 billion kWh, accounting for 7.2% of the total power generation, which can support the stable development of the energy base.

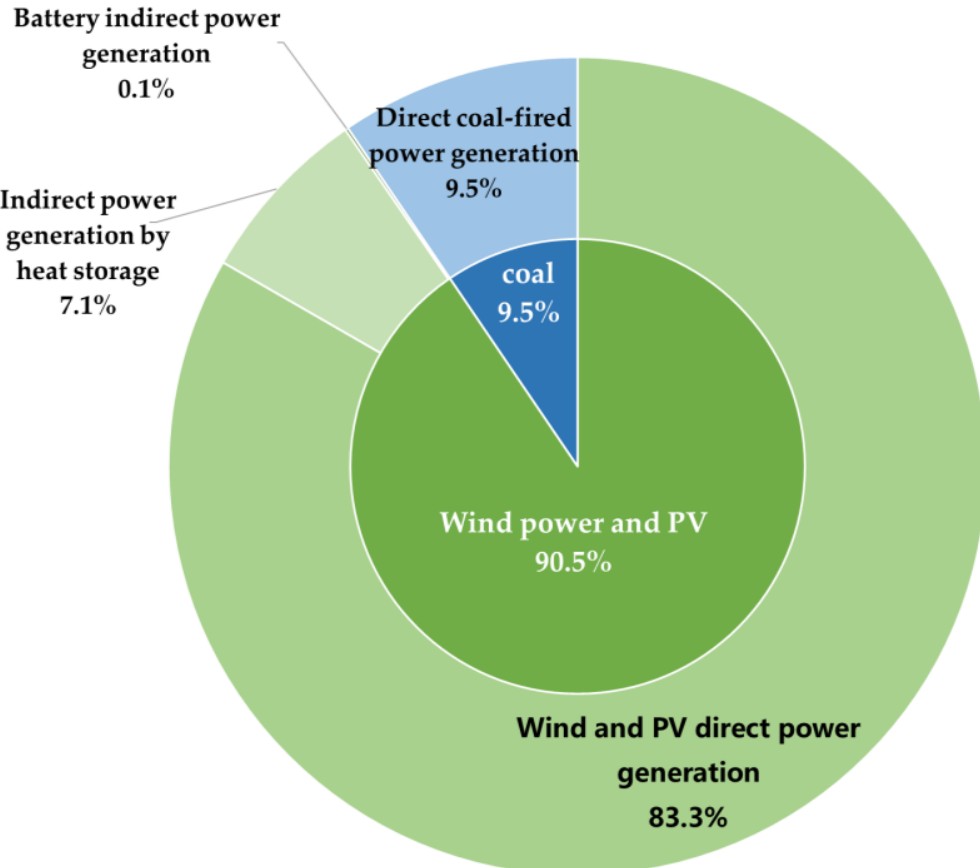

**Figure 14.** Power generation composition of a clean energy base.

### 4.4.2. Economic Analysis

1.    Optimize configuration

By adopting the above-optimized configuration, the demand for electrochemical energy storage configuration will be greatly reduced, and the total investment in the energy storage system can be effectively reduced (for energy storage systems with the same adjustment capacity, the investment will be reduced by more than 95%), thus ensuring the overall income of the energy base. From the annual power generation situation of the energy base, wind power and photovoltaic power generation account for 90.5% of the total power generation of the base so as to realize clean energy power generation to the maximum extent and reduce carbon emissions.

2.    Energy storage and multi-form power supply operation strategy

(1) Investment calculation

Prepare project investment calculations with reference to the regulations, cost quotas, and rate standards of the current relevant documents of the country, industry, and region where a clean energy base is located. The prices of main materials and equipment used in DC transmission, wind farms, photovoltaic, thermal power, heating, and other projects are uniformly estimated according to the 2020 price level. The overall investment frame of the clean energy base is shown in Table 1.

**Table 1.** Investment estimation of clean energy base.

| Engineering Subject | Scale | Unit Investment | Total Investment (Billion RMB Yuan) |
|---|---|---|---|
| UHVDC transmission | Self-build and send to a certain province, about 200 km. | — | 350.00 |
| | The power grid company builds, and the base pays the wheeling cost | — | 0 |
| Wind power | 10,000 MW | 6000 Yuan/kW | 600.00 |
| PV | 2000 MW | 4000 Yuan/kW | 80.00 |
| Thermal power | 4 × 1000 MW | 3512 Yuan/kW | 140.48 |
| Thermal storage | 80,000 MWh reserves | 4000 Yuan/kW | 30.00 |
| Battery energy storage power station | 150 MWh | 1300 Yuan/kW | 1.90 |
| Clean energy heating | 100,000 m$^2$ of heating area | — | 0.15 |
| Hydrogen production from wind power (hydrogenation) | 2000 standard cubic/h | — | 1.00 |
| Total investment | | Self-build | 1203.53 |
| | | The power grid company build | 853.53 |

The clean energy base has planned to build a new ±800 kV HHVDC transmission project, which will be delivered to a certain province, extending along the highway, with a distance of 1800–2000 km and a cost of about 35 billion yuan. The transmission line is

constructed by the power grid company, and the base needs to pay the wheeling cost (UHV transmission plus collection cost is about 0.09 Yuan/kWh).

10,000 MW wind power will be newly built. The static investment per kilowatt is 6000 yuan/kW, with a total investment of about 60 billion yuan; 2000 MW photovoltaic power will be newly built, the static investment per kilowatt is 4000 yuan/kW, and the total investment is about 8 billion yuan; 4000 MW thermal power will be newly built, including four 1000 MW units, with a static investment of 3512 yuan/kW per unit kilowatt, and the total investment is about 14.048 billion yuan; 80,000 MWh (reserves) thermal storage capacity will be Newly built, with a unit cost of 4000 yuan/ton and the total investment is about 3 billion yuan; a 50 MWh independent battery energy storage power station will be newly built, the unit kilowatt-hour investment of the battery energy storage system is 1300 yuan/kWh, with a total investment of about 190 million yuan. The energy base plans to build a clean energy heating system that can meet the heating load of 100,000 square meters system, with a total investment of nearly 15 million yuan. A new wind power hydrogen production (hydrogen refueling) station is planned to achieve a hydrogen production scale of 2000 standard cubic meters per hour, with a total investment of 100 million yuan.

According to the above investment calculation results, the prices of main materials and equipment used in wind power, PV, thermal power, DC transmission, and heating projects involved in the energy base are uniformly estimated according to the 2019 price level. If the base builds its own transmission lines, the overall static investment of the base will be 120.353 billion yuan. If the power grid company builds outbound transmission lines and the base pays the wheeling cost, the overall static investment of the base will be 85.353 billion yuan.

The investment in the energy base is mainly used for the construction and operation of wind power, photovoltaic, thermal power, UHV, DC transmission, battery energy storage, and heating projects in the base, and the primary source of revenue stems from electricity generation activities. In conservative estimation, revenues such as external heating and hydrogen output are not included for the time being; that is, the heating project also serves the base, and its surroundings is only a part of the investment without considering the benefits. The energy base has planned a hydrogen production project with a capacity of 2000 cubic meters/hour. As a part of the base investment, the hydrogen production income is not included in the economic calculation. In addition, the battery energy storage system is used for peak regulation and frequency regulation of the base power supply, which is only used as part of the investment.

(2) Economic feasibility calculation

Comprehensively considering indicators such as wind, PV, thermal power installed capacity, annual operating hours, and plant power consumption rate, the average construction period of the base is set to 3 years and put into production in batches. After full production, based on the installed capacity of 1600 million kilowatts, the annual average operating hours are converted to 2357.3 h, and the commissioning and operation period is uniformly calculated on the basis of 25 years.

The financial evaluation indicators of clean energy are shown in Tables 2 and 3. Using the overall economic calculation of the energy base, it is found that the base investment has a good internal rate of return, namely:

Scheme of self-built delivery lines from the base to a certain province: 8.00% (before tax) and 6.83% (after tax). When the power grid company builds an outgoing line to a certain province, the base pays the wheeling cost plan: 8.52% (before tax) and 7.3% (after tax).

The after-tax internal rate of return of the investment in the energy base has reached or exceeded the 8% standard generally adopted by the industry, which is worth investing in. At the same time, by comparing the two schemes, it is recommended to cooperate with the power grid company. The power grid company will build the transmission line, and the base will pay a certain wheeling cost, which can improve the profitability of the base.

**Table 2.** List of main indicators for the financial evaluation of an energy base project (self-build delivery line of the base to a province).

| Indexes | Value | Indexes | Value |
|---|---|---|---|
| Static investment in engineering | 120.353 billion yuan | Unit investment (converted) | 7522.06 Yuan/kW |
| Dynamic investment in engineering | 1236.61 billion yuan | Unit investment (converted) | 7728.81 Yuan/kW |
| Interest incurred during construction | 28.28 billion yuan | Working capital | 4.80 billion yuan |
| Electricity price (tax excluded) | 326.55 Yuan/MWh | Electricity price (tax included) | 369.00 Yuan/MWh |
| Financial internal rate of return on capital | 6.68% | Internal rate of return of bank finance | 6.84% |
| Total return on investment ROI | 5.53% | Net profit rate of capital ROE | 9.65% |
| Cash flow analysis of project investment | | | |
| | Internal rate of return (%) | Net present value (billion yuan) | Investment pay-back period (year) |
| Before income tax | 8 | 108.23 | 11.86 |
| After income tax | 6.83 | 90.33 | 12.71 |

**Table 3.** List of main indicators for the financial evaluation of an energy base project (the power grid company builds delivery lines to a province, and the base pays the network fee).

| Indexes | Value | Indexes | Value |
|---|---|---|---|
| Static investment in engineering | 853.53 billion yuan | Unit investment (converted) | 8334.56 Yuan/kW |
| Dynamic investment in engineering | 878.38 billion yuan | Unit investment (converted) | 5489.88 Yuan/kW |
| Interest incurred during construction | 20.05 billion yuan | Working capital | 4.80 billion yuan |
| Electricity price (tax excluded) | 246.90 Yuan/MWh | Electricity price (tax included) | 279.00 Yuan/MWh |
| Financial internal rate of return on capital | 7.61% | Internal rate of return of bank finance | 7.55% |
| Total return on investment ROI | 5.99% | Net profit rate of capital ROE | 10.95% |
| Cash flow analysis of project investment | | | |
| | Internal rate of return (%) | Net present value (billion yuan) | Investment pay-back period (year) |
| Before income tax | 8.52 | 118.17 | 11.41 |
| After income tax | 7.30 | 101.33 | 12.24 |

## 5. Conclusions

This paper systematically investigated the development of the energy storage industry, studied the development planning and system optimization configuration of base-based energy storage technology, and considered the constraints and indexes. The conclusions are as follows:

(1)  A new idea for base system optimization and operation scheduling strategy research is proposed

This paper put forward a new idea that a "thermal energy storage system based on molten salt heat storage, coupled with thermal power to undertake the main regulation task, and the battery energy storage system undertakes the frequency regulation task, taking into account the heating demand of the district". Propose the design idea of "giving priority to designing the main electric energy connection transmission network according to the energy form, then adding connection lines according to the energy storage order, and finally optimizing the load distribution network according to the load demand". The coordinated operation mode of "new energy priority grid connection, deep coupling between thermal power and thermal storage and fine adjustment of flexible energy storage" is put forward.

(2)  The optimal configuration of the base is completed, and the thermal storage and electricity storage are combined for operation

According to the new idea put forward in this paper, the optimal configuration scheme of energy storage and multi-form power sources is 10 million kilowatts for wind power, 2 million kilowatts for PV power, and $4 \times 1$ million kilowatts for thermal power. After comprehensive optimization calculation, it is necessary to allocate 80 million kWh of molten salt storage capacity, which can solve the problem of 85.4% power limitation. At the same time, it is necessary to allocate an average of 144,000 kW (1h) of electrochemical energy storage, which can flexibly solve the fine adjustment required by the energy base.

(3)  The economic feasibility evaluation is completed

Adopting the above-mentioned optimized configuration will greatly reduce the demand for electrochemical energy storage configuration, and for an energy storage system with the same adjustment capability, the investment will be reduced by more than 95%. Using the proposed operation scheduling strategy, the base has good economic benefits, and the overall pre-tax internal rate of return of the project reaches 8%.

**Author Contributions:** Methodology, M.L, Y.Z. and B.Z.; software, Y.Z. and B.Z.; Furthermore, J.W. (Jiaqi Wang); supervision, M.L.; validation, J.W. (Jiaqi Wang), J.W. (Jianxing Wang) and C.L.; visualization, J.Z., Y.S. and R.Z.; writing—original draft, M.L., Y.Z. and B.Z.; writing—review and editing, H.L., Y.S. and J.Z. All authors have read and agreed to the published version of the manuscript.

**Funding:** This study was supported by the Huaneng Group Headquarters Science and Technology Project of the Key Technology Research and System Development of Group Level Intelligence Operations Platform Construction (HNKJ21-H52) and the Fault Diagnosis Technology Research and System Development of Lithium-ion Battery Energy Storage Station Based on Mass Data (HNKJ21-H52-004).

**Institutional Review Board Statement:** Not applicable.

**Informed Consent Statement:** Not applicable.

**Data Availability Statement:** The datasets used during the current study are available from the corresponding author upon reasonable request.

**Conflicts of Interest:** The authors declare no conflict of interest.

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
