# Peer review of "Optimal Configuration of Wind-PV and Energy Storage in Large Clean Energy Bases"

_sustainability, doi:10.3390/su151712895_

Round 1

Reviewer 1 Report

  1. Authors are suggested to change the sentence formation of Abstract first line.
  2. For section 2 please add more reference and mention accountable outcomes from the reference only.
  3. For Constraints and Indexes please mention references from which they have been adopted.
  4. Add more recent references
  5. Authors should include the result justification with the considered constraints and indexes.
  6. Wind and solar power generation variation must be considered with separate case along with energy storage support.
  7.  Require some results showing the demand compensation with solar and wind for different variations.

Throughout the paper proofreading must be done.

Author Response

Please refer to the attachment for specific reply.

Reviewer 2 Report

The topic of the manuscript is very interesting, and some data are important for the design of actual low-carbon energy system. However, some place needs to be improved.

1. Distribution diagrame of active power output in time helps to illustrate characteristics of the wind power, and it should be illustated.

2. The design principles of thermal energy storage capacity, battery energy storage, and hydrogen production capacity should be expressed clearly. How to optimze capacity?

3. Does the the financial benefit analysis consider constrcution cost, installment cost, electric controll cost and maintenance cost?

4. Some sentences need to be improved.

Some sentences need to be improved.

Author Response

(The authors gave the same response as above.)

Reviewer 3 Report

The paper seeks to design an optimal configuration of thermal energy storage based on molten salt coupled with thermal power as well as wind-PV energy storage in large clean energy base. Renewable Energy storage is a very important subject in the global effort to combat climate change. This is a very important research area.

Find below my general comments about the paper.

1. Abstract:- The abstract must include a brief design of method.

2. Introduction:- The research gap is not clearly defined in the introduction.

Line 14:- Provide stable output..........there is something missing.

-incomplete statement.......Break through the operational data barriers of wind power, PV etc. 

The organisation of the paper is weak. There should be a theory and method sections which will incorporate the contents in sections 2-4.

Section 4.1 System modelling

-------There should be a preamble after system modelling-----

Section 4.1.1 Wind power generation modelling.

----insert a preamble-------

Fig 4 needs labelling.

Section 4.3:- refer to fig.12 in the discussion of 5,000 to 5,300 MW.

Page 12.......... and its system modules shown in fig 5 (remove the following figure and insert fig.5)

Section 4.4.1 (3rd line) . It can be seen---- insert i.

Fig 6 needs further explanation.

Fig 7 is missing-----

Results and Discussion

N/A

Conclusions

N/A

The English Language needs to improve in some parts of the paper.

Author Response

(The authors gave the same response as above.)

Reviewer 4 Report

mentioned within the attached document. 

Average English, it can be improved. 

Author Response

(The authors gave the same response as above.)

Round 2

Reviewer 1 Report

1. Authors suggested to do the proof reading.

Minor changes required

Author Response

Dear Reviewer,

Reviewer 3 Report

Abstract: The following sentence must be constructed properly:-

It is suggested that the base optimized with the power grid company, and the company will build the transmission line, and the base will pay a certain wheeling cost, which can further enhance the profitability of the base. 

The authors must create sections for theory and methodology

Section 3.2.1

It can be found from the Figure 2 that.... remove the.

The English Language looks ok except a few errors indicated above.

Author Response

Dear Reviewer,
